# Traits of Developmental Disorders in Adults With Listening Difficulties Without Diagnosis of Autism Spectrum Disorder And/or Attention-Deficit/Hyperactivity Disorder

**DOI:** 10.3390/jcm13206281

**Published:** 2024-10-21

**Authors:** Chie Obuchi, Tetsuaki Kawase, Yuka Sasame, Yayoi Yamamoto, Kaori Sasaki, Junya Iwasaki, Hidehiko Okamoto, Kimitaka Kaga

**Affiliations:** 1Institute of Human Sciences, University of Tsukuba, Tsukuba 305-8572, Japan; 2Tohoku University, Sendai 980-8577, Japan; kawase@orl.med.tohoku.ac.jp; 3Department of Speech and Hearing Sciences, International University of Health and Welfare, Chiba 286-8686, Japan; yyasuda@iuhw.ac.jp (Y.S.); yyamamoto@iuhw.ac.jp (Y.Y.); kaori.s@iuhw.ac.jp (K.S.); jiwasaki@iuhw.ac.jp (J.I.); 4Department of Physiology, International University of Health and Welfare, Chiba 286-8686, Japan; okamoto@iuhw.ac.jp; 5National Hospital Organization Tokyo Medical Center, National Institute of Sensory Organs, Tokyo 152-8902, Japan; kimitaka.kaga@kankakuki.jp

**Keywords:** listening difficulties, trait of developmental disorder, questionnaires, attention switching

## Abstract

**Background:** Some individuals have a normal audiogram but have listening difficulties (LiD). As many studies have investigated the relationship between listening and developmental disorders, the traits of developmental disorders might explain the symptoms of LiD. In this study, we examined the traits of developmental disorders of adults with LiD to help clarify the cause of LiD symptoms. **Methods:** In total, 60 adults with LiD and 57 adults without LiD were included. Participants completed a questionnaire for the autism spectrum quotient (AQ) test, the Adult Attention-Deficit Hyperactivity Disorder Self-Rating Scale (A-ADHD), the Adolescent/Adult Sensory Profile (SP), and the severity of subjective LiD in daily life. **Results:** Before analysis, we excluded participants with LiD who were already diagnosed or met the criteria for autism spectrum disorder (ASD) or ADHD, and the results of the remaining 30 participants (50.0%) with LiD were analyzed. Adults with LiD showed higher scores than those without LiD in the AQ. Attention switching in the AQ and attention ability in the A-ADHD scale were correlated with the severity of LiD symptoms in everyday life. The AQ scores were also significantly correlated with subscales of the SP. **Conclusions:** Adults with LiD showed greater autistic traits than those without LiD; therefore, LiD symptoms are possibly related to autistic symptoms. Furthermore, adults with LiD might have attention disorder traits of both ASD and ADHD and sensory processing problems. These findings suggest that the attention problems in adults with LiD noted in previous studies might be related to these traits of developmental disorders.

## 1. Introduction

### 1.1. Listening Difficulties and Their Causes

People with normal audiograms can generally hear sounds without difficulty. However, some individuals with a normal peripheral auditory system who are able to hear sounds struggle to listen to spoken messages. These challenges are amplified in noisy environments and when speech is unclear. These symptoms are commonly referred to as auditory processing disorders (APDs), and management methods have been summarized in guidelines from different countries [1,2]. These guidelines suggest that impaired auditory performance in individuals with a “normal” pure-tone audiogram may be due to disordered processing in the central auditory system. However, Moore [3] insists that other impairments must also be considered. He explains that impaired hair cell function and the resulting changes in cochlear compression influence spectral and temporal tuning independently of pure tone and top-down influences on listening, such as attention, memory, emotion, and learning. Furthermore, he recommends using the term “listening difficulties” (LiD) to describe the symptoms more appropriately. According to Moore [3], it is important to differentiate between problems associated with the peripheral auditory versus cognitive systems and to examine the specific issues related to listening difficulties. Furthermore, Moore et al. [4] noted that the concept of LiD recognizes multifactorial underlying causes that are not necessarily due to auditory system dysfunction. In this paper, we also use the term LiD to describe the symptoms of individuals with normal hearing and listening difficulties.

In the peripheral auditory system, cochlear synaptopathy is thought to cause similar listening difficulties in noisy situations and affect auditory perception in tasks. Previous studies on cochlear synaptopathy have reported that acoustic overexposure causes only transient threshold elevation and no hair cell loss; nevertheless, it can cause the irreversible loss of synapses between the inner hair cells and cochlear nerve fibers [5,6]. For the differential diagnosis of cochlear synaptopathy, noninvasive assessments, such as auditory brainstem responses, middle-ear muscle reflex, envelope-following responses, and extended high-frequency audiograms, are conducted [7]. However, these auditory assessments are affected by extraneous factors other than synaptopathy, and there are currently no clear and efficient evaluation methods. When examining people with LiD, it is important to inquire about their risk of acoustic exposure, and the likelihood of cochlear synaptopathy should be considered.

Besides these peripheral function problems, we consider the cognitive system to be the cause of LiD. Recent studies have reported that fundamental LiD is associated with cognitive weakness or poor attention [3,4,8,9,10,11,12,13,14,15] and speech-language processing mechanisms beyond the central auditory nervous system [4]. In these studies, children or adults with and without LiD conducted cognitive tasks. These studies found that people with LiD showed cognitive difficulties with challenging cognitive tasks. For instance, Petley et al. [12] also conducted a questionnaire (Evaluation of Children’s Listening and Processing Skills), performed a standardized clinical test suite for auditory processing, and administered cognition tests to children with and without LiD. They reported that children with LiD have problems with the cognitive processing of auditory and nonauditory stimuli that include both fluid and crystallized reasoning. In other words, LiD symptoms are associated with a variety of cognitive problems and have also been verified through evoked responses [16]. The LiD of people without peripheral auditory system disorders are attributed to their inability to differentiate important information from a large amount of input information or to maintain their attention to spoken messages.

Posner [17] divided the attention system into three subsystems: orienting toward sensory events, detecting signals for focal processing, and maintaining a vigilant or alert state. These attentional functions have since been shown to have four dimensions: selective, alternating, focused, and divided [18]. Auditory attention is important when listening to spoken messages. We hear and listen to various sounds using various attentional dimensions. If we divert our attention slightly from the message, we will be unable to comprehend the content. The role of attention in the background of LiD is well understood; however, the reason for this poor attention is unclear. Although some people with LiD have developmental disorders [19], to the best of our knowledge, the relationship between LiD and traits of developmental disorders has not been made sufficiently clear. 

### 1.2. Listening Difficulties and Traits of Developmental Disorders 

The relationship between LiD and developmental disorders, such as attention-deficit hyperactivity disorder (ADHD), dyslexia, learning disorder (LD), specific language impairment (SLI), and autism spectrum disorder (ASD), has been examined extensively. Dawes and Bishop [20] emphasized that auditory processing problems are one of several deficits commonly found in individuals with developmental disorders from an early stage. The British Society of Audiology [21] also reported that the high co-occurrence of APD with other developmental disorders in children, including SLI, dyslexia, and ASD, is widely recognized. De Wit et al. [22] conducted a systematic review of studies published in peer reviewed journals from 1954 to 2015 to investigate the similarities between APD and SLI, ADHD, LD, or ASD, as well as to identify characteristics that can distinguish children with APD from children with other developmental disorders. They showed that children with APD perform similarly to children with SLI, dyslexia, ADHD, and LD on tests of intelligence, memory or attention, and language tests, and only small differences between groups were found for sensory and perceptual functioning tasks. Levy and Parkin [23] considered that APD may be the root disorder underlying LD, ADHD, and SLI. Dillon et al. [24] also presented a taxonomy indicating that listening difficulties can be caused by deficits in multiple domains: APD, hearing deficits, language deficits, and cognitive deficits.

Jones et al. [25] examined the auditory processing difficulties identified among children with neurodevelopmental disorders, most notably in those with developmental language disorders and dyslexia, using computational simulation involving deep convolutional neural networks. They showed that the primary maturational frequency discrimination deficit may be associated with early language development. Haake et al. [26] assessed the basic auditory processing, phonological representation of word stress, and segmental contrasts for children with and without specific language impairment (SLI). They reported that children with SLI showed lower scores than those without SLI, and two different subtypes of impaired word stress processing emerged, an acoustic and a representational one. They emphasized the crucial importance of word stress processing for the successful acquisition of a broad variety of linguistic abilities (e.g., lexical, morphological, and syntactic). These findings showed that primary auditory perceptual deficits might be a crucial factor causing the language disorder. Auditory perception is important for language development, and it is essential to consider the relationship between the two. However, many people have LiD without any language developmental delay; the LiD symptoms cannot be explained from the perspective of language disorders.

Some research has noted a relationship between various types of listening problems and developmental disorders. For example, both children and adults with ASD exhibit auditory processing difficulties [27], and speech reception thresholds under background noise for high-functioning people with autism or Asperger’s syndrome were generally worse than those for the controls [28]. For individuals with dyslexia, speech-related auditory processing deficits may persist into adulthood, and auditory processing deficits are often seen in adults with dyslexia [29]. 

Regarding ADHD, Bench et al. [30] reported that most children referred for an auditory processing assessment showed characteristics of ADHD, a comorbidity of people diagnosed with both LiD and ADD. Another study [31] reported that LiD and ADHD may have similar symptoms; however, they are separate, largely independent conditions. As with these studies, there is still no consensus on the relationship between LiD and ADHD. Considering that the primary symptom of ADHD is attention deficits [32,33], people with ADHD might have difficulties orienting and maintaining their attention on the required information, and it would be difficult to divide them into two separate problems. 

Regarding listening in people with ASD, Bonnel et al. [34] reported superior performance in pitch discrimination and categorization; however, Kwakye et al. [35] noted that the auditory temporal thresholds of children with ASD were higher than those of typically developing children, showing impairment in auditory temporal processing in ASD. James et al. [36] showed that most school-age children with autism (86%) performed abnormally on at least one auditory test, suggesting that functional auditory problems can exist in people with autism despite normal pure-tone sensitivity. Balasco et al. [37] explained that the perceptual capabilities of people with ASD may be influenced by the nature and complexity of the sensory stimuli, with impairments associated with more complex stimuli and enhancements seen more often with simple stimuli. 

Furthermore, studies have demonstrated the effectiveness of a remote assistive device in children with ASD who have poor speech recognition [38,39]. The stress associated with listening was reduced using these assistive devices [40]. However, these studies were unable to explain why children with ASD have difficulty listening. 

These findings indicated a relationship between listening difficulties and developmental disorders. However, previous studies found that not all people with LiD had developmental problems [22,23]. It is possible that people with LiD have traits of an undiagnosed developmental disorder; however, this has not been proven. In particular, adults with LiD performed better than children with LiD in questionnaires and behavioral tests [27], and adults might not show obvious symptoms of developmental disorders. Therefore, it is important to examine the traits of developmental disorders in adults with LiD and the cause of LiD symptoms.

### 1.3. Research Question

As many studies have reported the relationship between listening and developmental disorders, the traits of developmental disorders might be one of the causes to explain the symptom of LiD. However, no previous studies have analyzed these traits in adults with LiD. Therefore, this study was designed to examine the traits of developmental disorders of adults with LiD using questionnaires that assess developmental disorders. The following research questions were addressed:(1)Are there any differences in traits of developmental disorders between adults with and without LiD?—Our hypothesis is that there are differences in the traits of developmental disorders between the two groups.(2)If there are differences among the participants, what are the characteristics of adults with LiD?—We hypothesize that the participants with LiD will show attention deficit tendencies such as ADHD and autistic tendencies such as ASD.

## 2. Materials and Methods

### 2.1. Participants

In total, 60 adults (mean age 21.9 ± 3.9 years; 19 men, 41 women), who had normal hearing but complained of LiD during everyday life, were enrolled in the study, and 57 participants (mean age 21.4 ± 0.6 years; 9 men, 45 women) who had normal hearing, no history of otology, and no complaints of LiD during everyday life were included as a control group. There was no significant age difference between the participating groups (t = 0.72, *p* = 0.47).

Adults with LiD were recruited from a peer support organization for people with LiD in Japan, and data were collected between April 2021 and February 2024. They were screened for normal hearing using standard audiometry at 125, 250, 500, 1000, 2000, 4000, and 8000 Hz, with bilateral thresholds of <20 dB HL. The participants’ extended high-frequency thresholds (10, 12, 14, and 16 kHz) were assessed using an audiometer (H-1, RION Co Ltd., Tokyo, Japan) with DD 450 headphones (DD450, RadioEar). All participants had normal extended high-frequency thresholds (<20 dB HL). Their syllable intelligibilities in the silent condition were normal; the average right ear score was 96.1% (standard deviation [SD]: 3.8), and the average left ear score was 95.7% (SD: 3.9). It was also confirmed using otoacoustic emission (OAE) or auditory brainstem response (ABR) that they had no hearing loss, and they had no otorhinolaryngological abnormalities on physical examination. The OAEs were measured with a clinical tool (Titan, Interacoustics, Middelfart, Denmark), and we confirmed the presence of DPOAE in each ear. The wave V of click-evoked ABR was measured using an evoked potential testing system (Neuropack, Nohon Kohden, Irvine, CA, USA), and we confirmed that the wave V was present in the participants.

The adults without LiD were university students. Their hearing thresholds were < 20 dB HL bilaterally, their speech intelligibilities were above 96%, and they had no problems with hearing loss using otoacoustic emissions. Interviews confirmed that the participants had no long-term exposure to loud noises or music. 

Participants’ subjective experiences with LiD were assessed using a questionnaire for adults with LiD [41]. This questionnaire was developed based on the findings of previous research [42]. They administered three questionnaires that were developed to assess the hearing status of people with hearing impairments, in contrast to adults with LiD. The questionnaire results for adults with LiD were significantly worse than those of the normal group, and there were strong correlations between the results of the auditory processing tests and the scores on the three questionnaires. Obuchi and Kaga [41] used one of the questionnaires and the shortened versions, the Speech, Spatial, and Qualities of Hearing Scale (SSQ-12; [43]), and combined four items from the psychological domain for the purpose of assessing participants’ psychological and social reactions to LiD. The final questionnaire consisted of 12 SSQ-12 items and 4 items from the psychological domain. The items had a scale score ranging from 0 to 10, corresponding to responses such as “not at all” and “perfect.” The total scores ranged from 0 to 160. Previous studies [41] specified that the cutoff value for LiD in this questionnaire was ≤109. The mean total score of participants with LiD was 74.9 (SD: 20.3; range 31–108) in this study, and their total scores were thus below the cutoff. The mean total score of participants without LiD was 130.7 (SD: 14.0), and the scores were above the cutoff.

Furthermore, participants with LiD underwent the Japanese auditory processing test (APT-J), a standardized test developed to assess the symptoms of LiD in children and adults with LiD in Japan (Gakuensya, Japan; [44,45]). This test is based on the traditional LiD assessment widely used abroad and consists of six subtests: dichotic listening test, natural fast speech perception test, gap detection test, speech-in-noise test, speech perception in multiple talker tests, and auditory simple attention test. All participants with LiD scored poorly on these subtests. Based on the above clinical results, the participants were diagnosed with or suspected to have LiD.

We excluded participants who were already diagnosed with ASD or ADHD to examine the traits of developmental disorders in adults with LiD who were not diagnosed with these disorders. Among the participants with LiD, four (6.7%) were diagnosed with ASD, six (10.0%) were diagnosed with ADHD, and two (3.3%) were diagnosed with ASD and ADHD. Among the participants without LiD, none were already diagnosed with ASD or ADHD. Finally, 48 adults with LiD and 57 adults without LiD participated in this study.

This study was approved by the ethics committee of the International University of Health and Welfare in Chiba Prefecture, Japan (reference number: #2020-Io-32), and was performed in accordance with the guidelines of the Declaration of Helsinki. All participants provided informed consent and were briefed in detail before the start of the study.

### 2.2. Tasks and Procedures

Participants completed a questionnaire for the autism spectrum quotient test (AQ), and we examined the relationship to the severity of the subjective experiences of LiD in daily life.

The AQ is a standardized questionnaire used to assess the ASD traits of participants. It is the most widely used screening questionnaire for children and adults with ASD in clinical situations in Japan. It was developed and validated by Baron-Cohen et al. [46] and translated into Japanese by Wakabayashi et al. [47]. It comprises five subscales: social skills (e.g., I prefer to do things with others rather than on my own), attention switching (e.g., I frequently get so strongly absorbed in one thing that I lose sight of other things), local attention (e.g., I often notice small sounds when others do not), communication (e.g., I frequently find that I don’t know how to keep a conversation going), and imagination (e.g., I don’t particularly enjoy reading fiction). The question items concerned autistic symptom traits and cognitive abnormalities in autism [46]. Each subscale consisted of 10 questions, and there were 50 questions in total. If the participants answered that they had abnormal or autistic-like behavior and responded to each question item, 1 point was added. The maximum score was 50, and the cutoff score of adults for ASD was 33. Previous studies have indicated that the AQ can differentiate between ASD, social anxiety disorders, obsessive compulsive disorder [48], and schizophrenia [49]. 

The severities of subjective experiences with LiD were assessed using a questionnaire for adults with LiD [41] as the criteria for participant selection. This questionnaire consisted of the SSQ-12 and four items of the psychological domain for assessing the psychological and social reactions to listening difficulties. We examined the relationship between traits of developmental disorders and severity of subjective experiences with LiD.

To analyze more detailed traits of developmental disorders, we conducted two additional developmental questionnaires, the Adult ADHD Self-Rating Scale (A-ADHD) and Adolescent/Adult Sensory Profile (SP) for participants with LiD. The A-ADHD was created in accordance with the Diagnostic and Statistical Manual of Mental Disorders, Fifth Edition, by Fukunishi in 2016 (Chiba test center). The questionnaire was widely used in various settings, including medical diagnosis and educational assessment in Japan. Twenty questionnaire items were formulated using symptoms related to the three major components: distractibility, hyperactivity, and impulsivity. Each question item was rated on a scale of 1 (I have none of the symptoms mentioned in the questionnaire) to 4 points (I always have the symptoms mentioned in the questionnaire). The maximum score was 80, and the cut-offs for ADHD were 54 for males and 52 for females. 

The SP was developed to measure adult responses to daily sensory experiences [50]. The questionnaire included four patterns of sensory processing described in Dunn’s model [51]: low registration (e.g., I do not notice when other people come in the room), sensation seeking (e.g., I like to attend events with a lot of noise), sensory sensitivity (e.g., I startle easily from unexpected or loud noises), and sensation avoiding (e.g., I avoid situations where unexpected things might happen). The total items were 60 with 15 items for each quadrant. Participants respond to each item with a 5-point Likert scale (i.e., 1 = always, 100% of the time; 2 = frequently, 75% of the time; 3 = occasionally, 50% of the time; 4 = seldom, 25% of the time; and 5 = never, 0% of the time). For the SP questionnaire, we examined the results for the 19 participants who were able to complete it.

Each participant completed these questionnaires, and the individual scores were calculated. 

### 2.3. Statistics

We calculated the mean and SD of the AQ subscales for each participant and compared the results statistically using a two-way analysis of variance (ANOVA) with the Scheffé test for post hoc comparisons when the ANOVA indicated significance. Regarding the subjective LiD questionnaire scores, we divided the questionnaire score into two domains, the shortened SSQ score and the psychological domain score. We analyzed the relationship between the scores of the AQ subscales and the scores of the LiD questionnaire using Pearson’s correlation for each participant group. 

Furthermore, for adults with LiD, the relationship between the AQ scores, LiD questionnaire scores and other developmental questionnaire scores, and the A-ADHD and sensory profiles was also analyzed using Pearson’s correlation.

All statistical analyses were conducted using the bell curve in Excel v.2.00 (Social Survey Research Information Co., Ltd., Tokyo, Japan). Differences were considered statistically significant at *p* < 0.05.

## 3. Results

During the research process, we excluded participants who met the criteria for ASD on the AQ or ADHD on the A-ADHD. Fourteen (23.3%) of the participants met the criteria for ASD, and four (6.7%) of the participants met the criteria for ADHD. The results of the remaining 30 participants (50.0%) with LiD were analyzed. There were 13 men and 17 women with a mean age of 21.4 years (SD: 3.9). Their mean hearing level of the right ear was 9.5 dB (SD: 6.5), and their mean hearing level of the left ear was 8.0dB (SD: 6.2). The average right-ear score was 95.6% (SD: 4.7), and the left-ear score was 95.4% (SD: 4.3). The mean total score of the LiD questionnaire in the participants with LiD was 78.7 (SD: 36.2; range 38–108).

The results for the mean and SD of the AQ subscale score for adults with and without LiD are shown in Figure 1. A two-way ANOVA showed the main effects of participants (F = 37.747, *p* < 0.001) and subscales (F = 16.086, *p* < 0.001). There was no interaction between participants and subscales (F = 1.624, *p* = 0.167). Post hoc analysis using Scheffe’s multiple comparison showed significant differences between social skills and communication (*p* < 0.001), social skills and imagination (*p* < 0.001), attention switching and attention to detail (*p* = 0.020), attention switching and communication (*p* < 0.001), attention switching and imagination (*p* < 0.001), and attention to detail and imagination (*p* = 0.004). 

The correlation table between the AQ subscale scores and LiD questionnaire scores for each participant group is presented in Table 1. For adults without LiD, the attention switching score was significantly correlated with the LiD questionnaire scores (SSQ score: r = −0.384, *p* = 0.003; psychological score: r = −0.320, *p* = 0.015). The SSQ scores, assessing the severity of listening difficulties in everyday life, were only correlated with the attention switching score of the AQ. The psychological score of the LiD questionnaire was also correlated with social skills and communication (social skills: r = −0.280, *p* = 0.029; communication: r = −0.384, *p* = 0.003). For adults with LiD, the attention switching score was significantly correlated with the SSQ scores (r = −0.382, *p* = 0.040). The psychological score of the LiD questionnaire was correlated with social skills and communication (social skills: r = −0.398, *p* = 0.029; communication: r = −0.462, *p* = 0.010). The relationships between the subscales of the AQ and LiD questionnaire that were significant in both groups are shown in Figure 2 as a scatter plot. The combined results of both adults with and without LiD showed significant correlations between the attention switching of the AQ and SSQ scores (r = −0.465, *p* < 0.001), psychological score and social skills of the AQ (r = −0.331, *p* = 0.002), and psychological score and communication of the AQ (r = −0.587, *p* < 0.001).

In addition, we analyzed the relationship between the AQ scores and A-ADHD scores in adults with LiD. The results revealed that the A-ADHD scores were significantly correlated between the AQ attention switching scores (r = 0.645, *p* < 0.001) and communication scores (r = 0.559, *p* < 0.001) (Figure 3). There were no significant correlations between the A-ADHD scores and social skills (r = 0.332, *p* = 0.073), attention to detail (r = −0.077, *p* = 0.686), or imagination (r = 0.135, *p* = 0.479).

Table 2 shows the relationship between the AQ scores, LiD questionnaire, and SP. Regarding the relationship between the AQ subscale scores and SP, there were significant correlations between many subscales of the AQ and SP. Regarding the psychological domain of the LiD questionnaire, it was significantly correlated with sensory sensitivity (r = −0.493, *p* = 0.049) and sensation avoiding (r = −0.564, *p* = 0.018).

## 4. Discussion

In this study, we examined the traits of developmental disorders in adults with and without LiD using several developmental questionnaires. Adults with LiD showed higher autistic traits than those without LiD. Differences were observed in all five subscales of the AQ: social skills, attention switching, attention to detail, communication, and imagination. In the selection of participants, half of the participants with LiD were excluded from this study because they were already diagnosed with ASD or ADHD or met the criteria for them on questionnaires. Although the remaining participants analyzed in this study had no diagnosis of ASD and did not meet the criteria, they had more autistic traits than those without LiD. These findings indicate the possibility that LiD symptoms are related to autistic symptoms. 

Recently, no clear distinction has been made between people with and without ASD, and the symptoms are on a spectrum [52]. This is the so-called “gray-zone” for people who are not clearly diagnosed with developmental disorders but have traits of developmental disorders. The diversity of profiles in individuals with ASD stem from either the presence of comorbid factors, as a core symptom of autistic behavior without comorbidity or both, with the development of complex clinical symptoms (Vogindroukas et al., 2022). The traits may be strong in some cases and weaker in others. It is possible that some adults with LiD in this study had gray-zone developmental disorders.

Considering each subscale score in the AQ, the SSQ scores in the LiD questionnaire, the severity of listening difficulties in everyday life, were correlated with attention switching in the AQ. The attention switching score showed the severity of specific attention problems based on ASD. Previous studies reported that people with ASD have impaired disengagement and orienting of attention, overly focused and narrow attention, and decreased ability to filter distractors [53,54]. Keehn et al. [55] reported that individuals with ASD often overlook salient behaviorally relevant information in their environment (e.g., their name being called or a person entering a room) but may appear to be distracted by subtle behaviorally irrelevant details within their surroundings (e.g., light shining through blinds or air flowing through a duct). The authors believed that the over-focus, combined with the susceptibility to distractions, among people with ASD may be due to dysfunctional modulation and interaction of attentional networks. Previous studies have also indicated attention problems in people with ASD, for example, selective attention or attention shift [56,57], attention-specific deficit in resistance to distractor inhibition [58], and increased processing of distractors [59]. Conversely, other studies have reported superior attention abilities [60,61]. However, people with ASD are thought to process irrelevant and potentially distracting information [59]. If adults with LiD have specificity of attention, they might be unable to pay attention to and listen to the information they need. They would also have difficulties switching their attention efficiently from one person’s message to another and listening to the message they need because they are distracted by non-essential noise around them. 

Attention switching scores in the AQ were also significantly correlated with the A-ADHD scores. This result indicates that the attention switching problem, as measured by the AQ, is related to inattention, as measured by the A-ADHD, although the quality is different. Hofvander et al. [62] assessed the autistic symptomatology, patterns of comorbid psychopathology, and psychosocial outcomes in 122 consecutively referred adults with normal-intelligence ASD. They reported that in 43% of adults diagnosed with ADHD, ASD and ADHD often co-occurred. Ames and White [63] indicated that most parents reported ADHD-related behaviors in children with a diagnosis of ASD, and 87% of children crossed the threshold for at least one component of ADHD. Polderman et al. [64] also investigated the co-occurrence between ASD and ADHD traits and their etiology. They showed that the co-occurrence of ADHD traits and autistic traits in adults is not determined by problems with hyperactivity, social skills, imagination, or routine preferences. Instead, the association between those traits is due primarily to shared attention-related problems, inattention, and attentional switching capacity. There are many other similar studies [13,65,66,67], and these previous studies showed findings with similar results to this study. Although participants with a history of ASD or ADHD diagnosis were strictly excluded from this study, adults with LiD might have both attention traits and show their difficulties in both questionnaires. The poor attention problems in adults with LiD noted in previous studies [3,4,8,9,10,11,12,13,14,15] might be related to these traits of developmental disorders. 

There were also significant correlations between attention switching in the AQ and SSQ scores, and social skills or communication and psychological scores of the LiD questionnaire in adults without LiD. As shown in Figure 2, the trend was the same when combined with the LiD results. Some adults without LiD showed similar results to those of adults with LiD, suggesting that adults with and without LiD might have a continuum of symptoms.

The psychological scores in the LiD questionnaire were correlated with social skills and communication in the AQ. Listening is related to communication and social skills, and people with LiD are unable to listen to spoken messages and communicate fluently with others. Therefore, they might avoid interacting with others, meaning that their social skills will also be lacking. They might have difficulties in communication and social interaction psychologically. Similar problems have been reported in people with hearing loss, such as a lack of social communication skills [68], and pragmatic language abilities [69]. Many people with ASD also experience problems with social skills and communication [70,71]. Their communication issues vary on a continuum of severity so that some children may be verbal, whereas others remain non-verbal or minimally verbal [71]. Vogindroukas et al. also explained that the diversity of profiles in speech and language development stem from either the presence of comorbid factors, as a core symptom of autistic behavior without comorbidity or both, with the development of complex clinical symptoms. As mentioned earlier, we thought that autistic traits were also involved in the correlation between social skills and communication.

These AQ scores were also significantly correlated with the scores of sensory sensitivity or sensation avoiding of the SP. As these sensory processing traits in ASD include both hypo- and hypersensory sensitivities [72], and the sensory symptoms were inversely related to mental age [73], a relationship is assumed between the two scales. There are many research reports on auditory hypersensitivity because it is conceptually easier to explain and because it is a more important clinical symptom than hypo-sensitivity [74]. The lifetime prevalence of auditory hypersensitivity in ASD was estimated to be 50.37 to 69.76% [75]. Lucker [76] measured the tolerable sound range for children with and without ASD and showed that auditory hypersensitivity is not based in the auditory system but is rather a conditioned response to sounds perceived as aversive or annoying. Furthermore, Lucker and Doman [77] reported that the primary neural mechanisms in auditory hypersensitivity involve negative emotional reactions to sound via connections between the auditory system and the limbic system as well as the limbic system and other parts of the body. The treatment for these hypersensitive hearing problems is felt to focus on changing the negative emotional reactions to more neutral ones and, hopefully, positive responses. If the LiD symptom is related to the auditory hypo- and hypersensitivity seen in ASD, cognitive therapy will probably be necessary to manage the symptom.

Robertson and Simmons [78] investigated whether the sensory difficulties experienced by those with ASD might extend to those with high levels of autistic traits in the general population by using questionnaires for sensory symptoms and the AQ. The results showed that autistic traits are continuously distributed in the general population, and there was a high correlation between the AQ score and the frequency of experiencing problematic sensory responses. If, as Robertson and Simmons [78] reported, the sensory differences experienced by adults with a diagnosis of ASD can also be extended into the general broad population and could be indicative of a sensory phenotype, adults with LiD in this study might also have sensory problems, and the sensory processing problems might cause the LiD symptoms.

The sensory processing abnormalities people with ASD show are explained by over- and under-responsiveness to sensory stimuli and failure to habituate [79,80]. The adults with LiD in this study, who have difficulties listening in everyday life, tended to have sensory sensitivities or sensation avoiding. Some of the listening symptoms they complain of might be the over-perception of unnecessary information and unconscious avoidance of sound messages. As a result of this, they might be unable to listen to necessary information.

However, not all adults with LiD have these autistic traits or sensory processing problems. The AQ scores of adults with LiD varied greatly in this study, and autistic traits only do not explain all the causes of LiD symptoms. Because the developmental traits of people with ASD vary [80], the individual differences in their traits might be increased. On the other hand, the causes of LiD symptoms differ, and if we analyze their symptoms as a whole, we might not achieve consistent results. In future studies, we must clarify how to distinguish between the individual causes of LiD in different people and establish clear criteria for diagnosing LiD.

In this study, we examined listening difficulties in everyday life and the traits of developmental disorders in adults with and without LiD using developmental questionnaires. The results revealed that adults with LiD showed higher autistic traits than those without LiD. Furthermore, the attention switching scores in the AQ were related to the A-ADHD scores and SSQ scores in the LiD questionnaire, and the AQ scores were related to the sensory processing scores. However, this study had several limitations.

First, the sample size of the research subjects in this study was limited. The final number of participants was small because we excluded participants with a diagnosis of ASD or ADHD and those who met these criteria. We need to collect a larger population to examine the findings of this study. Moreover, we must include in the comparison adults with ASD with and without symptoms of LiD. This investigation would better elucidate the relationship between autistic traits and LiD symptoms. Second, we must identify autistic traits in adults with LiD using not only subjective questionnaires but also behavioral experiments. The subjective feelings of difficulties are not always consistent with behavioral problems. We need to consider the LiD symptoms from both perspectives. Third, we need to examine the statistically significant results using effect size. We believe that the reliability and validity of the results of this study will be improved if we calculate the effect size as well as the *p*-value. Lastly, we were unable to examine other factors causing LiD symptoms. Mental conditions could also be related to LiD symptoms. We need to analyze the individual differences in symptoms that adults with LiD show and understand their symptoms so that we can consider evidence-based interventions for them.

## 5. Conclusions

We examined the listening difficulties in everyday life and the traits of developmental disorders in adults with and without LiD using developmental questionnaires. The results revealed that the adults with LiD showed higher autistic traits than those without LiD, and there might be the possibility that LiD symptoms are related to autistic symptoms. Furthermore, adults with LiD might have attention disorder traits of both ASD and ADHD and sensory processing problems. It was suggested that the poor attention problems in adults with LiD noted in previous studies might be related to these traits of developmental disorders.

## Figures and Tables

**Figure 1 jcm-13-06281-f001:**
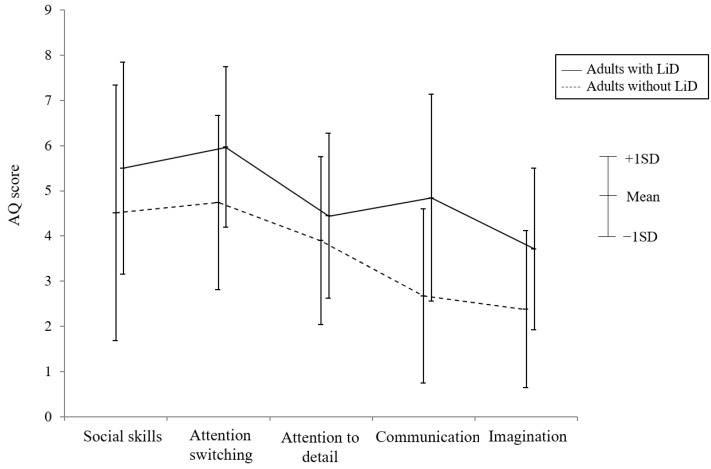
Comparison of the AQ subscale between adults with and without LiD. This figure presents the mean and standard deviation for each AQ subscale score for adults with and without LiD. A two-way ANOVA showed the main effects of the participants (F = 37.747, *p* < 0.001) and subscales (F = 16.086, *p* < 0.001). ANOVA: analysis of variance; AQ: autism spectrum quotient; LiD: listening difficulties.

**Figure 2 jcm-13-06281-f002:**
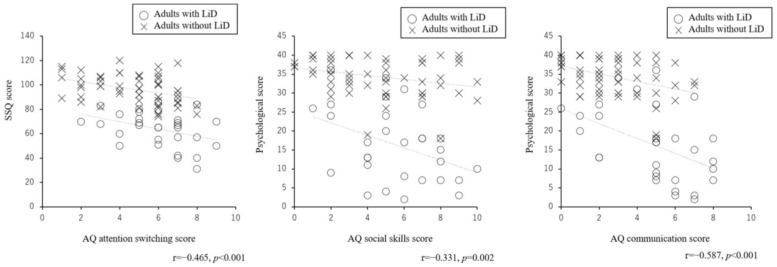
Scatter plot of the AQ subscale and LiD questionnaire in adults with and without LiD. This figure presents the relationships between the subscales of the AQ and LiD questionnaire that were significant in both groups as a scatter plot. The combined results of both adults with and without LiD showed significant correlations between the attention switching of the AQ and SSQ scores (r = −0.465, *p* < 0.001), psychological score and social skills of the AQ (r = −0.331, *p* = 0.002), and psychological score and communication of the AQ (r = −0.587, *p* < 0.001). AQ: autism spectrum quotient; LiD: listening difficulties; SSQ: Speech, Spatial, and Qualities of Hearing Scale.

**Figure 3 jcm-13-06281-f003:**
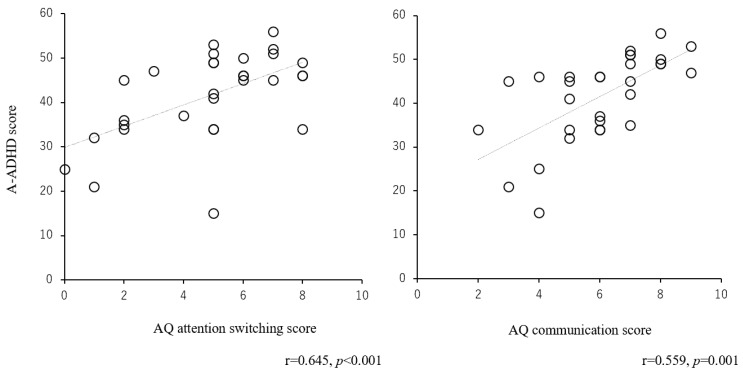
Relationship between the attention switching score or communication score of the AQ and A-ADHD score in people with LiD. This figure presents the relationship between the AQ scores and A-ADHD scores in adults with LiD. The results revealed that the A-ADHD scores were significantly correlated between the AQ attention switching scores and communication scores. There were no significant correlations between the A-ADHD scores and social skills, attention to detail, or imagination. A-ADHD: adult attention-deficit hyperactivity disorder; AQ: autism spectrum quotient; LiD: listening difficulties.

**Table 1 jcm-13-06281-t001:** Correlation table between AQ subscale scores and LiD questionnaire scores in adults with and without LiD.

Adults without LiD.					
	Social skills	Attention switching	Attention to detail	Communication	Imagination
SSQ score	−0.150	−0.384 **	0.075	−0.235	−0.066
Psychological score	−0.280 *	−0.320 **	0.012	−0.384 **	−0.097
Adults with LiD.					
	Social skills	Attention switching	Attention to detail	Communication	Imagination
SSQ	0.198	−0.382 *	−0.097	−0.149	−0.132
Psychological score	−0.398 *	−0.042	−0.106	−0.462 *	−0.082

* *p* < 0.05, ** *p* < 0.01. This table shows the relationship between the AQ scores and LiD questionnaire scores in adults with and without LiD. For adults without LiD, the attention switching score was significantly correlated with the LiD questionnaire scores. The SSQ scores, assessing the severity of listening difficulties in everyday life, were only correlated with the attention switching score of the AQ. The psychological score of the LiD questionnaire was also correlated with social skills and communication. For adults with LiD, the attention switching score was significantly correlated with the SSQ scores. The psychological score of the LiD questionnaire was correlated with social skills and communication. AQ: autism spectrum quotient; LiD: listening difficulties.

**Table 2 jcm-13-06281-t002:** Correlation table between AQ scores, LiD questionnaire and sensory processing in adults with LiD.

	AQ Score					LiD Questionnaire	
Sensory Profile	Social Skills	Attention Switching	Attention to Detail	Communication	Imagination	SSQ Score	Psychological Domain
Low registration	0.619 **	0.444	−0.273	0.685 **	0.410	−0.324	−0.376
Sensory seeking	−0.034	0.221	0.027	0.363	−0.010	−0.144	−0.170
Sensory sensitivity	0.443	0.676 **	0.004	0.398	0.344	−0.449	−0.493 *
Sensation avoiding	0.531 *	0.510 *	-0.001	0.617 **	0.481 *	−0.391	−0.564 *

* *p* < 0.05, ** *p* < 0.01. This table presents the relationships between the AQ scores, LiD questionnaire, and sensory processing in adults with LiD. The LiD questionnaire score was significantly correlated with sensory sensitivity and sensation avoiding. The psychological domain was significantly correlated with sensory sensitivity and sensation avoiding. Regarding the relationship between the SP and AQ scores, there were significant correlations between many subscales of the AQ and SP. AQ: autism spectrum quotient; LiD: listening difficulties.

## Data Availability

The data that support the findings of this study are available from the corresponding author upon reasonable request.

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
