# Peer review of "Traits of Developmental Disorders in Adults With Listening Difficulties Without Diagnosis of Autism Spectrum Disorder And/or Attention-Deficit/Hyperactivity Disorder"

_jcm, 2024, doi:10.3390/jcm13206281_

Round 1
Reviewer 1 Report
Comments and Suggestions for Authors This article is interesting and well written .The topic is relevant to the field and addresses the traits of developmental disorders in adults with listening difficulties It addresses a specific gap in this field as listening difficulties in presence of normal hearing is a widespread problem. This work adds more information to the subject area The conclusions re consistent with the evidence and arguments presented and they address the main question posed The references are appropriate.The tables and figures are explanatory
I have only few remarks:
Introduction
Line 98 The BSA [21] also reported..The acronym shoul be explained.. I have not found it.
Materials and methods
Lines 268-270 The questionnaire was included four patterns of sensory processing described in Dunn’s model [51]. I think there is a grammar mistake in this sentence
Author Response
We thank the reviewer for the time and effort dedicated to reviewing our manuscript. We sincerely appreciate the thoughtful suggestions and comments, which have greatly helped us to improve the manuscript. We have carefully considered these comments, responded to them below in a point-by-point manner, and revised the manuscript accordingly. All revisions are highlighted in yellow in the main manuscript. We trust that your comments have been addressed.
Comments: This article is interesting and well written .The topic is relevant to the field and addresses the traits of developmental disorders in adults with listening difficulties It addresses a specific gap in this field as listening difficulties in presence of normal hearing is a widespread problem. This work adds more information to the subject area The conclusions re consistent with the evidence and arguments presented and they address the main question posed The references are appropriate.The tables and figures are explanatory
I have only few remarks:
Introduction
Line 98 The BSA [21] also reported..The acronym shoul be explained.. I have not found it.
Materials and methods
Lines 268-270 The questionnaire was included four patterns of sensory processing described in Dunn’s model [51]. I think there is a grammar mistake in this sentence
Response: In accordance with your comment, we have showed full name of abbreviation and revised grammatical error.
Reviewer 2 Report
Comments and Suggestions for Authors
The subject is very interesting and actual as many people normal pure tone thresholds experience listening difficulties. At the same time, persons with autistic spectrum disorders and/or attention deficit/hyperactivity disoders can present traits similar to listening difficulties. So studies are needed to provide more detailed knowledge on this issue to personalize the management of such conditions.
The manuscript is of high quality in desing, writhing and presenting the results.
Just one comment on the number of participants:
of 60 persons with listening difficulties 12 were diagnosed with ASD and/or ADHD prior to the study and were excluded from analysis. So 48 must remain but in the text 45 were indicated (lines 224-227). Later in the results it has been pointed out that 18 persons were also excluded due to criteria of ADS and ADHD and total 30 participants with listening difficulties were finally included in the analysis (lines 293-296).
Author Response
We thank the reviewer for the time and effort dedicated to reviewing our manuscript. We sincerely appreciate the thoughtful suggestions and comments, which have greatly helped us to improve the manuscript. We have carefully considered these comments, responded to them below in a point-by-point manner, and revised the manuscript accordingly. All revisions are highlighted in yellow in the main manuscript. We trust that your comments have been addressed.
Comments:
The subject is very interesting and actual as many people normal pure tone thresholds experience listening difficulties. At the same time, persons with autistic spectrum disorders and/or attention deficit/hyperactivity disoders can present traits similar to listening difficulties. So studies are needed to provide more detailed knowledge on this issue to personalize the management of such conditions.
The manuscript is of high quality in desing, writhing and presenting the results.
Just one comment on the number of participants:
of 60 persons with listening difficulties 12 were diagnosed with ASD and/or ADHD prior to the study and were excluded from analysis. So 48 must remain but in the text 45 were indicated (lines 224-227). Later in the results it has been pointed out that 18 persons were also excluded due to criteria of ADS and ADHD and total 30 participants with listening difficulties were finally included in the analysis (lines 293-296).
Response: We apologize for our mistake. The number 48 is incorrect, the correct number is 45. We revised the number.
Reviewer 3 Report
Comments and Suggestions for Authors
Dear authors, dear Editor;
The present study aimed to evaluate the relationship between listening difficulties disease LID and traits of developmental disorders. The introduction section is sufficient with clearly described research questions. The MM is thorough to provide reproducibility. The results are presented in a good way, and the discussion is well written and the conclusions supported by the results.
Minor Comments:
- Please state your hypothesis after the research questions.
- I dont finf the post-hoc test results in the ANOVA analyses ? If dont, please provide post-hoc test results and describe which test you have used (Bonferroni, Tukey, LSD, etc).
- Please provide effect sizes for the statistical significant results.
- Include the ethical statement also in the MM section as seperate paragraph.
Author Response
We thank the reviewer for the time and effort dedicated to reviewing our manuscript. We sincerely appreciate the thoughtful suggestions and comments, which have greatly helped us to improve the manuscript. We have carefully considered these comments, responded to them below in a point-by-point manner, and revised the manuscript accordingly. All revisions are highlighted in yellow in the main manuscript. We trust that your comments have been addressed.
Comments: The present study aimed to evaluate the relationship between listening difficulties disease LID and traits of developmental disorders. The introduction section is sufficient with clearly described research questions. The MM is thorough to provide reproducibility. The results are presented in a good way, and the discussion is well written and the conclusions supported by the results.
Minor Comments:
1) Please state your hypothesis after the research questions.
Response: In accordance with your comment, we have added our hypotheses.
2) I dont finf the post-hoc test results in the ANOVA analyses ? If dont, please provide post-hoc test results and describe which test you have used (Bonferroni, Tukey, LSD, etc).
Response: We apologize for the confusing description. We have already included the post-hoc test results in the ANOVA (Line 318-322)
3) Please provide effect sizes for the statistical significant results.
Response: Thank you for your suggestion. We understand the importance of effect sizes. However, our statistical software we currently own is unable to analyze the effect size of all the statistical analyses we have conducted in this study. It will take some time to show the effect size. For this reason, we have added it as a future consideration.
4) Include the ethical statement also in the MM section as seperate paragraph.
Response: Thank you for your suggestion. We already included the ethical statement in main text (Line 239-242).
Reviewer 4 Report
Comments and Suggestions for Authors
I have read with interest your study. Few minor revision requested. First, you may follow STROBE guidelines to enhance the quality and reproducibility of the study. The study briefly mentions otoacoustic emissions and auditory brainstem response to confirm no hearing loss, but providing detailed information on the specific protocols followed (e.g., equipment used, criteria for normal results, and any calibration procedures) would enhance the reproducibility of the study. Reporting details of the included population after exclusions (such as age, sex, and any other relevant demographic or clinical characteristics) is critical for transparency and understanding the representativeness of the study. Providing this data in a clear and concise table format would improve clarity for readers and allow for better comparison with other studies.
Author Response
We thank the reviewer for the time and effort dedicated to reviewing our manuscript. We sincerely appreciate the thoughtful suggestions and comments, which have greatly helped us to improve the manuscript. We have carefully considered these comments, responded to them below in a point-by-point manner, and revised the manuscript accordingly. All revisions are highlighted in yellow in the main manuscript. We trust that your comments have been addressed.
Comments: I have read with interest your study. Few minor revision requested. First, you may follow STROBE guidelines to enhance the quality and reproducibility of the study. The study briefly mentions otoacoustic emissions and auditory brainstem response to confirm no hearing loss, but providing detailed information on the specific protocols followed (e.g., equipment used, criteria for normal results, and any calibration procedures) would enhance the reproducibility of the study. Reporting details of the included population after exclusions (such as age, sex, and any other relevant demographic or clinical characteristics) is critical for transparency and understanding the representativeness of the study. Providing this data in a clear and concise table format would improve clarity for readers and allow for better comparison with other studies.
Response: In accordance with your comment, we have added more detailed information about OAE and ABR. Furthermore, we have added the descriptions of population after exclusions. As we did not analyze the results of each individual, we described the information of participants in the main text, not in the table.